# Different Associations between Tonsil Microbiome, Chronic Tonsillitis, and Intermittent Hypoxemia among Obstructive Sleep Apnea Children of Different Weight Status: A Pilot Case-Control Study

**DOI:** 10.3390/jpm11060486

**Published:** 2021-05-28

**Authors:** Hai-Hua Chuang, Jen-Fu Hsu, Li-Pang Chuang, Cheng-Hsun Chiu, Yen-Lin Huang, Hsueh-Yu Li, Ning-Hung Chen, Yu-Shu Huang, Chun-Wei Chuang, Chung-Guei Huang, Hsin-Chih Lai, Li-Ang Lee

**Affiliations:** 1Department of Family Medicine, Chang Gung Memorial Hospital, Taipei Branch and Linkou Main Branch, Taoyuan 33305, Taiwan; chhaihua@cgmh.org.tw; 2College of Medicine, Chang Gung University, Taoyuan 33302, Taiwan; jeff0724@gmail.com (J.-F.H.); r5243@cgmh.org.tw (L.-P.C.); chchiu@cgmh.org.tw (C.-H.C.); dochempath@cgmh.org.tw (Y.-L.H.); hyli38@cgmh.org.tw (H.-Y.L.); nhchen@cgmh.org.tw (N.-H.C.); yushuhuang1212@gmail.com (Y.-S.H.); 3Department of Industrial Engineering and Management, National Taipei University of Technology, Taipei 10608, Taiwan; 4Obesity Institute, Genomic Medicine Institute, Geisinger, Danville, PA 17822, USA; 5Department of Pediatrics, Chang Gung Memorial Hospital, Linkou Main Branch, Taoyuan 33305, Taiwan; 6Department of Pulmonary and Critical Care Medicine, Chang Gung Memorial Hospital, Linkou Main Branch, Taoyuan 33305, Taiwan; 7Department of Pathology, Chang Gung Memorial Hospital, Linkou Main Branch, Taoyuan 33305, Taiwan; 8Department of Otorhinolaryngology-Head and Neck Surgery, Chang Gung Memorial Hospital, Linkou Main Branch, Taoyuan 33305, Taiwan; 9Department of Child Psychiatry, Chang Gung Memorial Hospital, Linkou Main Branch, Taoyuan 33305, Taiwan; 10Department of Laboratory Medicine, Chang Gung Memorial Hospital, Linkou Main Branch, Taoyuan 33305, Taiwan; whitereverie5336@gmail.com (C.-W.C.); joyce@cgmh.org.tw (C.-G.H.); hclai@mail.cgu.edu.tw (H.-C.L.); 11Department of Medical Biotechnology and Laboratory Science, Graduate Institute of Biomedical Sciences, Chang Gung University, Taoyuan 33302, Taiwan

**Keywords:** children, intermittent hypoxemia, microbiome, obstructive sleep apnea, tonsil, weight status

## Abstract

The tonsil microbiome is associated with chronic tonsillitis and obstructive sleep apnea (OSA) in children, and the gut microbiome is associated with host weight status. In this study, we hypothesized that weight status may be associated with clinical profiles and the tonsil microbiome in children with OSA. We prospectively enrolled 33 non-healthy-weight (cases) and 33 healthy-weight (controls) pediatric OSA patients matched by the proportion of chronic tonsillitis. Differences in the tonsil microbiome between the non-healthy-weight and healthy-weight subgroups and relationships between the tonsil microbiome and clinical variables were investigated. Non-healthy weight was associated with significant intermittent hypoxemia (oxygen desaturation index, mean blood saturation (SpO_2_), and minimal SpO_2_) and higher systolic blood pressure percentile, but was not related to the tonsil microbiome. However, chronic tonsillitis was related to Acidobacteria in the non-healthy-weight subgroup, and oxygen desaturation index was associated with Bacteroidetes in the healthy-weight subgroup. In post hoc analysis, the children with mean SpO_2_ ≤ 97% had reduced *α* and *β* diversities and a higher abundance of Bacteroidetes than those with mean SpO_2_ > 97%. These preliminary findings are novel and provide insights into future research to understand the pathogenesis of the disease and develop personalized treatments for pediatric OSA.

## 1. Introduction

Obstructive sleep apnea (OSA) is a chronic disorder characterized by intermittent partial or complete upper airway obstruction during sleep. The prevalence of pediatric OSA is estimated to be 1–4%, with adenotonsillar hypertrophy and overweight/obesity being the two most important risk factors [1,2,3]. Pediatric OSA is of great clinical significance since evidence has shown a wide range of detrimental long-term effects associated with the condition [1]. For example, children with OSA show higher risks of neurobehavioral impairment [4], metabolic alterations [5], and cardiovascular dysfunction [6].

The role of microbiota in the development and aggravation of OSA has gained increasing attention. Previous studies have reported that the gut microbiota is involved in the pathogenesis of OSA [7,8], obesity [9,10], and hypertension [11,12]. For example, the transplant of fecal microbiota has been shown to elicit sleep disturbance [8], obesity [13], and hypertension [12] in animal models. The gut microbiota has also been associated with intermittent hypoxia and systemic inflammation [7,10], which are both well-documented manifestations of OSA [14,15,16]. More recent studies have suggested that, in addition to the gut microbiota, OSA is linked to alterations in various other microbiomes in the human body such as the nasal cavity [17], adenoids [18], tonsils [19], oropharynx [20], oral cavity [21], lungs [22], and urine [21]. In children who snore, the adenotonsillar microbiome has been shown to interact with the regional mucosal immune system such as interleukin-8 and heat shock protein 27 [23].

Tonsil size is one of the most important predictors for apnea-hypopnea index (AHI) in preschoolers and school-age children [24]. Therefore, the influence of the tonsil microbiome may be significant on OSA in young children. Two main methods are used to detect bacterial communities on tonsils: swab cultures and culture-free molecular tests based on 16S ribosomal RNA or ribosomal DNA sequencing [25]. Notably, molecular tests enable metagenomic studies to better detect slow-growing, uncultivable, and rare bacteria [26]. The advent of metagenomics has led to an increase in investigations on human microbiota. However, studies on the tonsil microbiomes in pediatric OSA patients and their relationships with patient characteristics, disease severity, and hypertension are still lacking. To the best of our knowledge, the clinical significance of tonsil microbiota in children with OSA has not been comprehensively elucidated.

We hypothesized that the tonsil microbiome may be associated with the weight status and anthropometrics of pediatric OSA patients. Furthermore, the relationships between the tonsil microbiome, OSA severity, intermittent hypoxemia, and hypertension may differ across patients with various demographic and clinical parameters. Therefore, among a cohort of children with OSA, the first aim of this study was to investigate differences in the tonsil microbiome between non-healthy-weight and healthy-weight subgroups. The second aim was to perform post hoc analysis to understand the correlations between the tonsil microbiome and other variables of interest, including OSA severity, intermittent hypoxemia, and hypertension.

## 2. Materials and Methods

### 2.1. Ethical Considerations

This was a prospective case-control study. Consecutive pediatric patients referred to the Department of Otolaryngology at Chang Gung Memorial Hospital (Linkou Main Branch, Taoyuan, Taiwan) for adenotonsillectomy between 1 March 2017 and 31 January 2019 were recruited. The Institutional Review Board of Chang Gung Medical Foundation approved this study (201507279A3), and all procedures were conducted in compliance with the Declaration of Helsinki 1975. Written informed consent was obtained from all parents and participants ≥6 years of age.

### 2.2. Patient Selection and Grouping

All of the participants underwent comprehensive history-taking, physical examinations, and standard in-lab polysomnography (PSG). The protocol was previously published [15]. The inclusion criteria were: (1) age 5–12 years, and (2) AHI ≥ 5.0 events/h or AHI ≥ 2.0 events/h plus at least one morbidity (such as elevated blood pressure (BP), daytime sleepiness, learning problems, growth failure, or enuresis) [14,27,28]. Patients with craniofacial, neuromuscular, or chronic inflammatory disorders (such as atopic dermatitis, asthma, or autoimmune disease) were excluded [14,15]. The subjects were further divided into two subgroups according to body mass index (BMI) z-score: “non-healthy-weight” (≤−2.0 kg/m^2^ and ≥1.0 kg/m^2^) group, and “healthy-weight” (>−2.0 kg/m^2^ and <1.0 kg/m^2^) group [29]. Both groups were matched by the proportion of chronic tonsillitis. Chronic tonsillitis was defined as symptoms of tonsillitis that persisted for a period longer than three months [30]. Patients with acute inflammation, such as rhinosinusitis, tonsillitis, gastrointestinal infection, or other conditions that needed antibiotic treatment did not undergo surgery after the diseases diminished for at least 2 weeks [15]. Subjective OSA symptoms (evaluated using the Chinese version of the OSA-18 questionnaire [31,32]), tonsil size (rated using the Brodsky grading scale [33]), the adenoidal-nasopharyngeal ratio (ANR) (measured using lateral radiography of the nasopharynx [34]), and allergic rhinitis were recorded. Figure 1 shows the flow diagram of the study.

### 2.3. Polysomnography Variables

We assessed OSA severity variables (AHI, respiratory disturbance index (RDI), oxygen desaturation index (ODI), mean pulse oxygen saturation (SpO_2_), and minimal SpO_2_) by standard full-night, in-lab PSG, according to the 2012 American Academy of Sleep Medicine Manual [35]. Briefly, the AHI was defined as the sum of all apneas (≥90% decrease in airflow for a duration of ≥2 breaths) plus hypopneas (≥50% decrease in airflow and either ≥ 3% desaturation or electroencephalographic arousal, for a duration of ≥2 breaths) divided by the number of hours of total sleep time. The patients were then categorized as having either severe (obstructive AHI ≥ 10.0 events/h) or non-severe (obstructive AHI ≥ 2.0 events/h to < 10.0 events/h) OSA [36]. The RDI was defined as the average number of respiratory disturbances (obstructive apneas, hypopneas, and respiratory-event-related arousals) per hour. The ODI was calculated as the average number of respiratory events with a 3% drop in SpO_2_ per hour. Furthermore, sleep stages were scored from electroencephalography records according to conventional criteria. PSG scoring was performed by a technician blinded to the clinical status of the children. Details of the PSG protocol were described previously [15,37].

### 2.4. Nocturnal Blood Pressure

Before the PSG exam, nocturnal BP was measured three times with a standard sphygmomanometer between 10:00 and 11:00 PM. The detailed procedure of measuring BP is described elsewhere [38]. Age, sex, and height-corrected percentiles of systolic BP (SBP) and diastolic BP (DBP) were recorded for each child [39]. Pediatric hypertension was defined as an average clinic SBP and/or DBP ≥ 95th percentile [39].

### 2.5. Tonsil Microbiota

The current study chose tonsil as the primary site for investigation since previous research suggested that the adenoidal microbiome was compatible with the tonsillar microbiome at the phylum level [23]. Also, since bacterial colonies were mainly observed in the tonsil crypts rather than in the tonsil follicles, the superficial tonsils with crypts were used for molecular examinations.

Tonsils with crypts were excised using sterile scissors during adenotonsillectomy. The specimens were rinsed with normal saline to remove superficial debris several times after harvesting. Genomic DNA was immediately extracted from the superficially biopsied specimens (3 mm × 3 mm × 3 mm) using an EasyPrep Genomic DNA Extraction Kit (Biotools Co., Ltd., New Taipei, Taiwan). Tonsil tissue was treated with 4 μL of RNase A (100 mg/mL) for 5 min at room temperature followed by 20 μL of Proteinase K at 56 °C until completely lysed, and then 200 μL ethanol (96–100%) for 15 s [40]. The quality and quantity of genomic DNA were measured using a NanoPhotometer P360 system (Implen, Westlake Village, CA, USA). Polymerase chain reaction (PCR) was used to amplify the V3–V4 regions of the gene encoding for 16S rRNA in bacteria using composite primers, including the forward primer 5′-TCGTCGGCAGCGTCAGATGTGTATAAGAGACAGCCTAYGGGRBGCASCAG-3′ and the reverse primer 5′-GTCTCGTGGGCTCGGAGATGTGTATAAGAGACAGGGACTACNNGGGTATCTAAT-3′ [41]. Amplicons were purified using a QiaQuick PCR Purification Kit (Qiagen, Hilden, Germany). PCR amplicons were sequenced using the Illumina HiSeq 2500 platform (Illumina, Inc., San Diego, CA, USA) following the manufacturer’s instructions to generate 250 bp paired-end reads.

All of the paired-end reads were assembled using FLASH software (version 1.2.7; http://ccb.jhu.edu/software/FLASH/ (accessed on 5 February 2019)) [42], and reads with a quality score < 20 were removed using QIIME software (version 1.7; http://qiime.org/ (accessed on 5 February 2019)) [43]. Sequences were chimera-checked using UCHIME software (http://drive5.com/uchime/ (accessed on 5 February 2019)) [44] and filtered from the dataset before the operational taxonomic unit (OTU) picking of 97% sequence identity using USEARCH (version 7) [45]. Taxonomy classification was annotated according to the Greengenes database (version 13.8; http://greengenes.secondgenome.com/ (accessed on 5 February 2019)) [46]. Multiple sequences were aligned using PyNAST software (version 1.2; https://pypi.org/project/pynast/ (accessed on 5 February 2019)) against the Greengenes core set database to identify the relationships between different OTUs [47]. We used Graphical Phylogenetic Analysis to visualize microbial genomes and metagenomes [48]. Detailed protocols of bioinformatics were described previously [41,49]. During data collection and analysis, the investigators were blinded to group allocation.

### 2.6. Sample Size Estimation

The sample size was estimated using primary outcome effects (BMI z-score) based on a priori study criteria [15] (healthy-weight group = 0.52 ± 1.12 and non-healthy-weight group = 1.53 ± 1.05). We used a two-tailed Wilcoxon–Mann–Whitney test to calculate the sample size (effect size = 0.93; type I error = 0.05; power = 0.95), which generated a sample size of 33 in each group.

### 2.7. Statistical Analysis

The D’Agostino and Pearson normality test showed that most variables had non-normal distribution. Therefore, descriptive statistics were expressed as the median, interquartile range (IQR), or frequency. Differences in variables of interest between specific (weight, OSA severity, BP) subgroups were determined using the Mann–Whitney *U* test, Kruskal–Wallis test, or chi-square test as appropriate. Analysis of similarity was performed to compare bacterial communities [50]. The α diversity (the diversity within each sample) of the tonsil sample was calculated using the observed richness based on the frequency of OTUs and genera in the sequence collections [51]. The β diversity (the number of species shared between two groups) was calculated using the weighted UniFrac measure [52]. Spearman’s correlation test was used to determine associations among major (>0.1% abundance and present in >90% of samples [49]) or minor phyla with the patient characteristics. Overall taxonomic or phylum-level abundances were included when determining the most discriminatory taxa between the two groups. Statistical significance was established at *p* < 0.05. *p*-values were corrected for multiple comparisons using the Benjamini–Hochberg method at 0.1 and reported as *q*-values when appropriate [53]. All statistical analyses were conducted using R software (versions 2.15.3 and 3.6.1, R Foundation for Statistical Computing, Vienna, Austria; http://www.r-project.org/ (accessed on 26 February 2021)) and Graph Pad Prism software (version 9.00; Graph Pad Software Inc., San Diego, CA, USA).

## 3. Results

### 3.1. Participants’ Characteristics

Figure 1 demonstrates the study flow diagram. Seventy-six Taiwanese children of Han ancestry with OSA were assessed for eligibility, 10 of whom were excluded from this study. Therefore, a total of 66 children with OSA (16 girls and 50 boys; median age, 6.5 years (IQR, 6.0–9.0); median BMI, 17.1 kg/m^2^ (IQR, 15.2–22.7); median AHI, 8.5 events/hour (IQR, 4.1–19.5); median SBP, 103 mmHg (IQR, 95–114); median DBP, 64 mmHg (IQR, 58–71)) were enrolled.

The children were further divided into two subgroups according to BMI z-score: non-healthy-weight subgroup (cases; *n* = 33), and healthy-weight subgroup (controls; *n* = 33). In the non-healthy-weight subgroup, 30 (91%) children had a BMI z-score ≥ 1.0 kg/m^2^ and three (9%) had a BMI z-score ≤ −2.0 kg/m^2^. All participants underwent adenotonsillectomy and were included for primary statistical analysis. The median time interval between PSG and adenotonsillectomy was 1 week (IQR: 6–19 weeks).

### 3.2. Differences in Participants’ Characteristics, PSG Variables, BP, and Tonsil Microbiome between the Different Weight Status Subgroups

#### 3.2.1. Differences in Participants’ Characteristics, PSG Variables, and BP between the Different Weight Status Subgroups

As expected, there was no significant difference in the proportion of chronic tonsillitis between the two subgroups (Table 1). Furthermore, there were no statistically significant differences in the proportions of male sex, allergic rhinitis, tonsil size, ANR, OSA-18 score, AHI, RDI, N1 stage, N2 stage, N3 stage, rapid eye movement (REM) stage, and DBP percentile. Notably, the non-healthy-weight group had significantly higher age, BMI z-score, ODI, SBP, DBP, and SBP percentile, and lower mean SpO_2_ and minimal SpO_2_ than the healthy-weight group. The difference in the tonsil size between children with chronic tonsillitis (3 (IQR, 3–3)) and children without chronic tonsillitis (3 (IQR, 3–4)) did not reach a statistical significance (*p* = 0.13).

Furthermore, the differences in participants’ characteristics, PSG variables, and BP between the overweight and underweight subgroups were not statistically significant (Appendix A). These variables were also comparable across the overweight, underweight, and healthy-weight subgroups (all *p* > 0.05).

#### 3.2.2. Differences in Tonsil Microbiome between the Different Weight Status Subgroups

After quality assessment, a total of 4,207,400 16S rRNA paired-end reads with an average of 105,185 ± 19,957 paired-end reads per sample passed the filters. Figure 2A,B show OTU trees of the non-healthy-weight subgroup (OTU = 9318) and healthy-weight subgroup (OTU = 9886). Figure 2C shows that both subgroups shared 6539 OTUs; otherwise, there were 2779 and 3347 deferential OTUs in the non-healthy-weight and healthy-weight subgroups, respectively. However, there were no significant differences in *α* diversity, *β* diversity, and relative abundances of the top 10 tonsil families between the non-healthy-weight and healthy-weight subgroups (Figure 2D–F; all *p* > 0.05). Furthermore, the *α* diversity, *β* diversity, and relative abundance of the top 10 tonsil families of the over-weight were comparable with those of the underweight subgroups (*p* = 0.064, 0.106, 0.492, respectively). Moreover, the differences in *α* diversity, *β* diversity, and relative abundances of the top 10 tonsil families across the overweight, underweight, and healthy-weight subgroups (*p* = 0.088, 0.119, 0.700, respectively).

Fifty-five phyla were identified from the tonsil samples. There were six major phyla (>0.1% abundance and present in >90% of the samples [49]), including *Proteobacteria*, *Firmicutes*, *Bacteroidetes*, *Fusobacteria*, *Actinobacteria*, and *Epsilonbacteraeota*, and 49 minor phyla. In descending order of median relative abundance, the 10 most common phyla were *Proteobacteria*, *Firmicutes*, *Bacteroidetes*, *Fusobacteria*, *Actinobacteria*, *Epsilonbacteraeota*, *Patescibacteria*, *Cyanobacteria*, *Tenericutes*, and *Acidobacteria.*

Figure 3A,B demonstrate the similar distributions of the relative abundances of these phyla in the non-healthy-weight and healthy-weight subgroups (*p* > 0.05). Furthermore, the relative abundances of these phyla in the overweight and underweight subgroups were comparable (*p* = 0.536). Additionally, the differences in the 10 most common phyla across the overweight, underweight, and healthy-weight subgroups were not statistically significant (*p* = 0.814).

#### 3.2.3. Associations of Tonsil Phyla, Participants’ Characteristics, PSG Variables, and BPs in the Different Weight Status Subgroups

In the non-healthy-weight group, age was related to Cyanobacteria and Acidobacteria, chronic tonsillitis was correlated with Acidobacteria, and SBP percentile was associated with Firmicutes (Figure 3C). The positive relationship between chronic tonsillitis and Acidobacteria remained significant after applying the Benjamini–Hochberg method (*r* = 0.53, *q* = 0.015).

Although there were several weak associations between the PSG variables (AHI, RDI, ODI, mean SpO_2_, minimal SpO_2_), DBP percentile, and Bacteroidetes, ODI and Actinobacteria, stage 1 sleep and Firmicutes, stage 3 sleep and Tenericutes, and REM stage and Proteobacteria, only the significant positive association between ODI and Bacteroidetes persisted after applying the Benjamini–Hochberg method (*r* = 0.52, *q* = 0.020) (Figure 3D).

### 3.3. Post Hoc Analysis

After studying the differences between the two weight subgroups, we wondered whether other classifications may be associated with the tonsil microbiome. We performed median splits of the participants’ characteristics, PSG variables, and BPs. The *α* diversities of the tonsil microbiome were not associated with age ≥ 6 years, male sex, chronic tonsillitis, BMI z-score ≥ 1.00, AHI ≥ 9.0 events/h, RDI ≥ 9.0 events/h, ODI ≥ 5.0 events/h, minimal SpO_2_ ≤ 90%, N1 stage ≥ 10%, N2 stage ≥ 40%, N3 stage ≤ 26%, REM stage ≥ 19%, SBP percentile ≥ 70%, and DSBP percentile ≥ 70%. In addition, differences in the tonsil microbiome between various (moderate-to-severe OSA and mild OSA; severe OSA, and non-severe OSA; hypertension and non-hypertension) subgroups did not reach statistical significance.

Interestingly, both α (Figure 4A) and β (Figure 4B) diversity indices of the mean SpO_2_ ≤ 97% subgroup were significantly lower than those of the mean SpO_2_ > 97% subgroup (*p* = 0.030 and 0.0005, respectively). The relative abundances of the top 10 tonsil phyla in the mean SpO_2_ ≤ 97% subgroup were significantly different from those in the mean SpO_2_ > 97% subgroup (*p* = 0.014; analysis of similarity test) (Figure 4C). Notably, the relative abundance of Bacteroidetes in the mean SpO_2_ ≤ 97% subgroup was significantly higher than that in the mean SpO_2_ > 97% subgroup (q = 0.026) (Figure 4D).

## 4. Discussion

In the following paragraphs, we address several novel and interesting findings of this study regarding the relationships of the tonsil microbiome with weight status, OSA severity, and hypoxemia among a sample of pediatric OSA patients.

Some clinical parameters were significantly different between the non-healthy-weight and healthy-weight subgroups. The non-healthy-weight subgroup had more profound intermittent hypoxemia (ODI, mean SpO_2_, and minimal SpO_2_) and a higher SBP percentile, which is consistent with the well-known connections between obesity and manifestations of OSA. However, no significant association was observed between the tonsil microbiome and weight status. The data did not support our primary hypothesis. Notably, although weight status was not directly associated with the tonsil microbiome in the overall cohort, relationships between the tonsil microbiome and other clinical parameters differed between the patients with different weight statuses. Chronic tonsillitis was related to Acidobacteria in the non-healthy-weight subgroup, while ODI was associated with Bacteroidetes in the healthy-weight subgroup. In post hoc analysis, we found that the children with or without a mean SpO_2_ ≤ 97% had significantly different microbial profiles, especially with regards to Bacteroidetes. It seemed that, instead of weight status, hypoxemia status was the key differentiating factor for the tonsil microbiota among pediatric OSA patients.

Previous studies have suggested that the presence and severity of OSA are associated with microbial profiles [19,20,23]. Yang et al. used 16S ribosomal DNA sequencing to investigate the oropharyngeal microbiome and demonstrated that adults with OSA had less oropharyngeal species diversity and altered abundance compared with non-OSA controls, and that the relative abundance of *Neisseria* (a genus of Proteobacteria) increased with higher OSA severity [20]. In our results, neither Proteobacteria nor *Neisseria* was related to AHI (*p* = 0.905 and 0.246, respectively). However, the children with a mean SpO_2_ ≤ 97% had less tonsillar species diversity and altered abundance compared to those with a mean SpO_2_ > 97%. Again, this suggested that the degree of intermittent hypoxemia may be more influential on the tonsil microbiome than OSA severity.

Johnston et al. investigated the tonsillar crypt microbiota in children with recurrent tonsillitis [54] and OSA [19], and found that Fusobacteria, Proteobacteria, Bacteroidetes, and Firmicutes were major phyla of the tonsil specimens in both groups (*α* diversity: *p* = 0.66; *β* diversity: *p* = 0.52) [19]. In another study on the adenotonsillar microbiome of children who snored [23], Kim and colleagues also reported similar major phyla, namely Proteobacteria, Actinobacteria, Firmicutes, Fusobacteria, Bacteroidetes, and Tenericutes. They further demonstrated that *α* diversity indices were related to some patient characteristics such as sex, emotional stress, and interleukin-8, and that *β* diversity indices were related to heat shock protein 70. These findings suggest possible connections between demographic characteristics, clinical symptoms, regional mucosal immune environment, and tonsil microbiome.

Our data are compatible with the results of the previous studies concerning the general picture of the tonsil microbiome. Moreover, we found a positive correlation between chronic tonsillitis and Acidobacteria in the non-healthy-weight subgroup. To the best of our knowledge, this is the first study to suggest that Acidobacteria may be implicated in tonsillar infections among pediatric OSA patients. Acidobacteria are Gram-negative rod-shaped bacteria. The majority of Acidobacteria strains have been described as aerobes, and they are ubiquitous in soil [55]. An increasing number of studies have investigated Acidobacteria in humans. For example, Acidobacteria are reported to be the fifth most dominant phyla in the bronchoalveolar lavage of adults with OSA [22]. In addition, both Acidobacteria and obesity are associated with an increase in the fecal levels of valeric acid [56,57]. Furthermore, Acidobacteria are shown to be significantly enriched in patients with chronic endodontic infection [58]. These observations provide indirect evidence for possible connections between tonsillar Acidobacteria infection, OSA, and obesity, and future investigations are warranted to confirm the causality.

The other novel finding of our study is the significant association between intermittent hypoxemia and Bacteroidetes in the healthy-weight subgroup. Members of the phylum Bacteroidetes are Gram-negative, rod-shaped, anaerobic or aerobic bacteria, and they are commonly found in the oral cavity and gastrointestinal tract. The abundance of Bacteroidetes and the Bacteroidetes–Firmicutes ratio are reported to be decreased in obese individuals compared to lean individuals [59]. However, our data suggested that it was the healthy-weight pediatric OSA patients in whom ODI significantly interacted with Bacteroidetes. Bacteroidetes of the tonsils may thus be implicated in the pathophysiology of OSA.

Previous studies have suggested associations between OSA and alterations in the composition and diversity of fecal microbiota. In a murine model, intermittent hypoxia exposure led to a lower abundance of Bacteroidetes in the feces [60], and reintroduction of a normoxic environment did not reverse the negative alterations of the gut microbiota [61]. In a sample of adults with OSA, Ko et al. found that fecal *Bacteroides* (a major genus of Bacteroidetes) were not associated with AHI or intermittent hypoxemia [62]. However, the results were very different for the oral microbiome. Xu et al. reported a higher abundance of oral Bacteroidetes in children with OSA [21]. Consistent with this finding, we also found a higher abundance of Bacteroidetes in pediatric OSA patients with a lower mean SpO_2_. The discrepancy between fecal and oral microbiota as well as the differences between patients with and without profound hypoxia are very interesting and may be explained by the oxygen concentration level at different sites and the aerotolerant abilities of different bacteria.

The environments of sites in the human body impact which microorganisms can inhabit these sites. Unlike the environment of the intestine, which is extremely low in oxygen concentration [63], to survive in the oral cavity, bacteria need to overcome the challenge of atmospheric oxygen exposure. Moreover, mouth breathing is very common in children with OSA [64] and positively associated with AHI [65]. OSA-related mouth breathing further increases exposure of the tonsil microbiome to atmospheric oxygen. *Bacteroides* species are among the most aerotolerant anaerobes and are able to tolerate oxygen in room air for up to 3 days [66]. On the other hand, *Fusobacterium* species (a genus of Fusobacteria) and *Clostridium* species (a genus of Firmicutes) are less aerotolerant anaerobes. Intermittent hypoxemia (in terms of mean SpO_2_ ≤ 97%) further enhances the survival advantage of Bacteroidetes relative to other aerobes of the tonsils. Mouse models of chronic intermittent hypoxia would be helpful to further validate these inferences [60].

Several limitations should be addressed in this study. First, the patient number of the underweight subgroup was too small and insufficient to make a conclusion. Future studies with a larger sample size of each weight status subgroup are warranted to further understand how obesity or underweight may impact the tonsil microbiome and its interaction with OSA. Second, the study cohort was predominantly male and mostly Han in ethnicity, which may limit the generalizability of the results. Third, some of the children may have had co-existing chronic tonsillitis, which would interfere with the analysis of the tonsil microbiome. However, the proportions of sex and chronic tonsillitis were comparable in both weight subgroups to minimize confounding effects from baseline characteristics. Forth, the study was cross-sectional and thus unable to conclude the direction of associations or causal effects. The relationships between the microbiota and OSA need to be further explored. Also, future prospective investigations on the effects of OSA treatment on the tonsil microbiome with a larger sample size will be of interest.

## 5. Conclusions

The advent of metagenomics has led to an increase in investigations on human microbiota. The tonsil microbiome plays a role in pediatric OSA, and it seems to have different effects depending on weight status. We preliminarily found that chronic tonsillitis was related to Acidobacteria in children with OSA and non-healthy weight, and that ODI was associated with Bacteroidetes in the children with OSA and healthy weight. In addition, children with OSA with or without mean SpO_2_ ≤ 97% had significantly different microbial profiles, particularly with regards to Bacteroidetes. Future studies to investigate associations among alterations of the tonsil microbiome and exacerbations or reductions of OSA severity are warranted. Furthermore, this study also suggests the possibility of personalized treatment of pediatric OSA based on the tonsil microbiome.

## Figures and Tables

**Figure 1 jpm-11-00486-f001:**
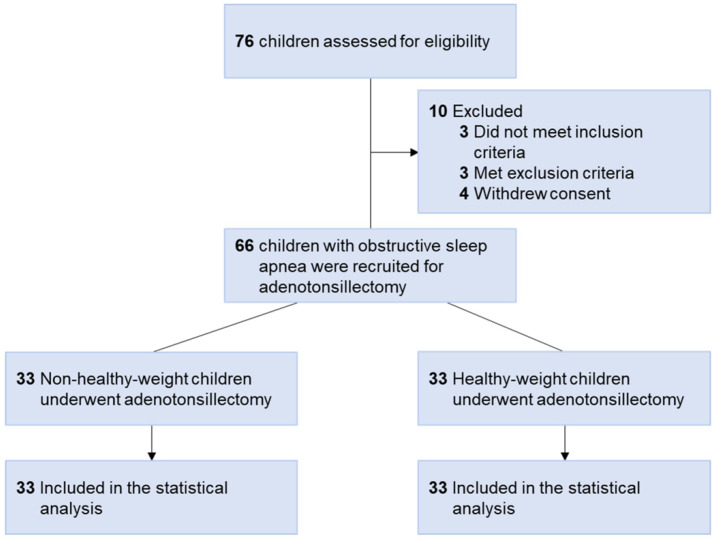
Flow diagram of the study. Seventy-six children with obstructive sleep apnea were assessed for eligibility. However, three did not meet the inclusion criteria, three met the exclusion criteria, and four withdrew consent. Therefore, a total of 66 children were recruited. The non-healthy-weight group included 33 children, and the healthy-weight group included 33 children. Both groups were matched by the proportion of chronic tonsillitis. All participants underwent adenotonsillectomy. Therefore, 66 participants were included in the primary analysis.

**Figure 2 jpm-11-00486-f002:**
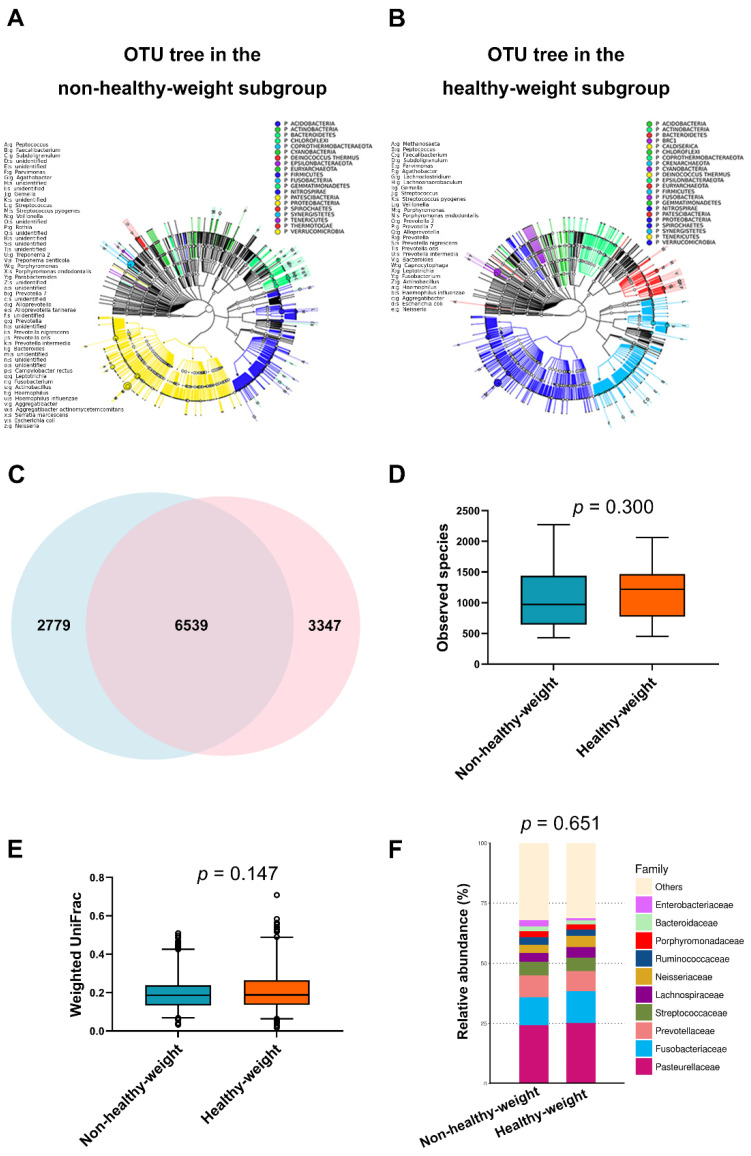
Tonsil microbiome in children with obstructive sleep apnea. (**A**) The operational taxonomic unit (OTU) tree of the non-healthy-weight subgroup (*n* = 33) included 9318 OTUs, assessed by Graphical Phylogenetic Analysis. (**B**) The OTU tree of the healthy-weight subgroup (*n* = 33) included 9886 OTUs. (**C**) A Venn diagram demonstrated that both subgroups shared 6539 OTUs; otherwise, the non-healthy-weight subgroup had 2779 deferential OTUs and the healthy-weight subgroup had 3347 deferential OTUs. (**D**) The *α* diversities of both subgroups were equal (*p* = 0.300; Mann–Whitney *U* test). (**E**) Furthermore, the *β* diversity of the non-healthy-weight subgroup was comparable to that of the healthy-weight group (*p* = 0.147; Mann–Whitney *U* test). (**F**) The relative abundances of the top 10 tonsil families in the non-healthy-weight subgroup were similar to those in the healthy-weight subgroup (*p* = 0.651; analysis of similarity test).

**Figure 3 jpm-11-00486-f003:**
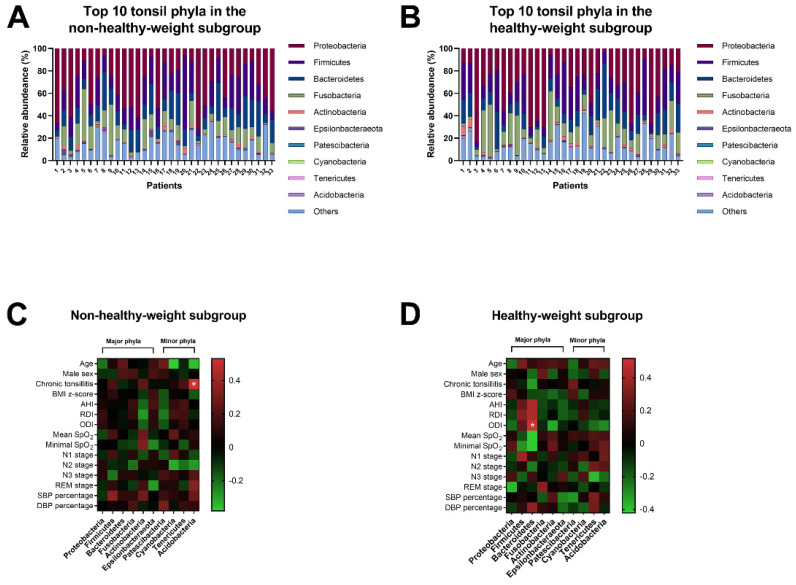
The top 10 tonsil phyla in both weight status subgroups. The relative abundances of the top 10 tonsil phyla in the non-healthy-weight subgroup (**A**) and the healthy-weight subgroup (**B**) (*p* = 0.651; analysis of similarity test). Chronic tonsillitis was significantly associated with a relative abundance of Acidobacteria in the non-healthy-weight subgroup (*r* = 0.53, *q* = 0.015) (**C**), whereas the oxygen desaturation index (ODI) was significantly associated with Bacteroidetes in the healthy-weight subgroup (*r* = 0.52, *q* = 0.020) (**D**). Abbreviations: AHI, apnea–hypopnea index; BMI, body mass index; RDI, respiratory disturbance index; REM, rapid eye movement; SpO_2_, pulse oxygen saturation.

**Figure 4 jpm-11-00486-f004:**
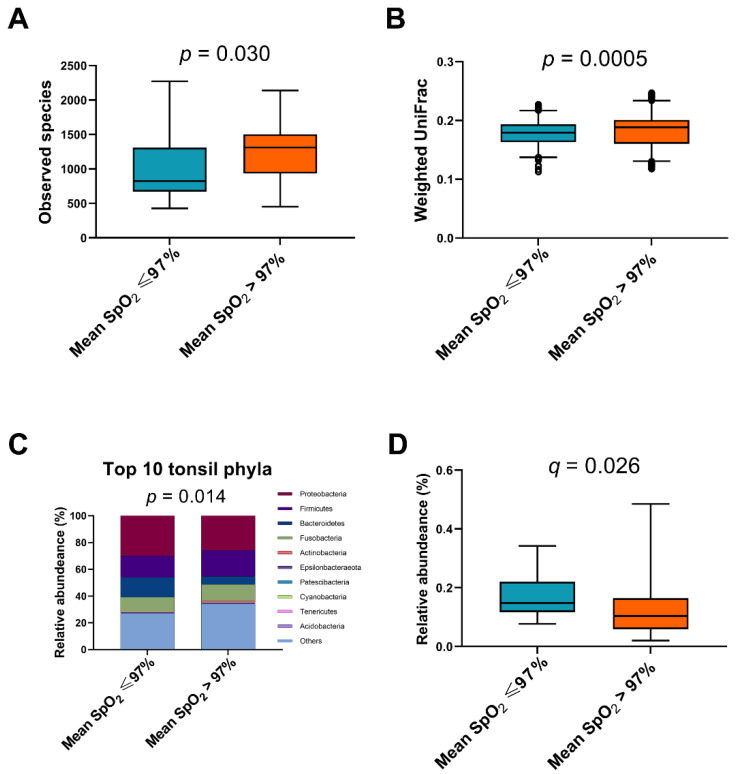
Comparison of the tonsil microbiome between the mean oxygen saturation (SpO_2_) ≤ 97% and > 97% subgroups. (**A**) The *α* diversity of the mean SpO_2_ ≤ 97% subgroup was significantly lower than that of the mean SpO_2_ > 97% subgroup (*p* = 0.030; Mann–Whitney *U* test). (**B**) The *β* diversity of the mean SpO_2_ ≤ 97% subgroup was significantly lower than that of the SpO_2_ > 97% subgroup (*p* = 0.0005; Mann–Whitney *U* test). (**C**) The relative abundances of the top 10 tonsil phyla in the mean SpO_2_ ≤ 97% subgroup were significantly different from those in the mean SpO_2_ > 97% subgroup (*p* = 0.014; analysis of similarity test). (**D**) Furthermore, the relative abundance of Bacteroidetes in the mean SpO_2_ ≤ 97% subgroup was significantly higher than that in the mean SpO_2_ > 97% subgroup (*q* = 0.026; Mann–Whitney *U* test).

**Table 1 jpm-11-00486-t001:** Patient characteristics, polysomnography variables, and blood pressures of the different weight status subgroups.

Variables	Non-Healthy-Weight Subgroup	Healthy-Weight Subgroup	*p*-Value ^1^
Patient Characteristics			
Age (years)	7.0 (6.0–8.0)	6.0 (5.0–7.5)	0.015 *
Male sex, *n* (%)	28 (85)	22 (67)	0.150
Chronic tonsillitis	6 (18)	10 (30)	0.389
Allergic rhinitis, *n* (%)	22 (67%)	25 (76%)	0.587
BMI (kg/m^2^) z-score	2.01 (1.46–2.38)	−0.36 (−1.16–0.18)	<0.001 *
Tonsil size	3 (3–4)	2 (3–4)	0.461
ANR	0.73 (0.62–0.83)	0.81 (0.72–0.87)	0.053
OSA-18 score	80 (69–92)	81 (70–91)	0.928
Polysomnography variables			
AHI (events/h)	9.6 (5.0–25.2)	5.4 (3.9–16.5)	0.074
RDI (events/h)	12.1 (5.3–27.6)	6.1 (4.9–17.9)	0.158
ODI (events/h)	7.3 (3.6–22.3)	3.2 (1.6–9.3)	0.006 *
Mean SpO_2_ (%)	97 (96–98)	98 (97–98)	0.030 *
Minimal SpO_2_ (%)	89 (83–91)	91 (88–93)	0.022 *
N1 stage	13 (6–21)	9 (6–13)	0.142
N2 stage	38 (33–46)	41 (36–44)	0.807
N3 stage	28 (23–30)	28 (22–36)	0.663
REM stage	18 (13–22)	21 (16–25)	0.221
Blood pressure variables			
Systolic BP, mmHg	111 (100–121)	98 (87–107)	0.001 *
Diastolic BP, mmHg	67 (61–76)	60 (58–68)	0.011 *
Systolic BP percentile (%)	84 (55–91)	48 (25–89)	0.018 *
Diastolic BP percentile (%)	75 (55–87)	68 (50–77)	0.174

Note: Data are summarized as median (interquartile range) or *n* (%) as appropriate. Abbreviations: AHI, apnea–hypopnea index; ANR, adenoid–nasopharyngeal ratio; BMI, body mass index; BP, blood pressure; ODI, oxygen desaturation index; OSA, obstructive sleep apnea; RDI, respiratory disturbance index; REM, rapid eye movement; SpO_2_, pulse oxygen saturation. ^1^ Data were compared using the Mann–Whitney U test for continuous variables, and the chi-square test for categorical variables. * Significant differences *p* < 0.05.

## Data Availability

The data presented in this study are available upon request from the corresponding author. The data are not publicly available due to ethical restrictions.

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
