# Peer review of "Different Associations between Tonsil Microbiome, Chronic Tonsillitis, and Intermittent Hypoxemia among Obstructive Sleep Apnea Children of Different Weight Status: A Pilot Case-Control Study"

_jpm, 2021, doi:10.3390/jpm11060486_

Round 1

Reviewer 1 Report

In this manuscript, authors evaluated the association between tonsil microbiome, chronic tonsillitis, and intermittent hypoxia in OSA children according to weight status. This is a prospective study, single center based, case-control study. Authors contain polysomnographic data of OSA children and evaluated microbiome in 67 children, which is relatively large number. The results are interesting and valuable. However, specific study design need to be compensated.

-Introduction

:In line 52, 67, authors described that ‘adenotonsillar’ hypertrophy and ‘adenotonsillar’ microbiome are important in OSA. However, from line 70, authors only comment on ‘tonsillar’ microbiome. Why did authors only consider tonsillar microbiome in OSA patients?

-Materials and Methods

:How was ‘chronic tonsillitis’ diagnosed?

:Why did authors include underweight children together with overweight children into a same group? The pathophysiology of overweight and underweight in OSA may be too different. They should be considered in a different group.

:How about the time schedule for polysomnography and tonsillectomy? (For example, polysomnography was performed at ---- weeks before surgery…)

:Information about medication history also should be described in study (including antibiotics).

:Not only ‘chronic inflammation’, but ‘acute inflammation’ such as acute rhinosinusitis is a confounding factor in this study, and should be evaluated based on objective evaluation such as PNS X-ray.

:Subjective discomfort could be also an important contributing factor. Did authors collect any data about subjective discomfort score from patients or parents?

:Regarding tissue sampling, ‘tonsils with crytps’ means superficial? or deep tissue biopsy? Microbiome between superficial area and deep tissue could be different, and tissue sampling is an important issue in microbiome study. And it would be better to add information about the size of samples.

-Results

: In table 1, information about tonsil size should be added. Also the difference of tonsil size between chronic tonsillitis and without chronic tonsillitis also should be evaluated. Size of tonsil and adenoid is an important contributing factor in hypoxia event of children.

: Authors grouped over-weight and under-weight patients into a ‘non-healthy weight subgroup’. How about the results when only over-weight or underweight patients were evaluated?

-Discussion

:Again, the results not between’ non-healthy’ and ‘healthy’ but between ‘overweight’ and ‘normal-weight’ or ‘underweight’ and ‘overweight’ might be more convincing, and need to be mentioned here.

:To further support authors suggestion (for examples, significant association between intermittent hypoxia and Bacteroidetes), chronic intermittent hypoxia mouse model would be helpful in further study.

:Allergic condition is an important factor in regional microbiome. If authors have data about allergic diseases such as allergic rhinitis, allergic asthma, it should be added in table 1 and it should be considered in data analysis.

Author Response

Reviewer 1’s comments:

In this manuscript, authors evaluated the association between tonsil microbiome, chronic tonsillitis, and intermittent hypoxia in OSA children according to weight status. This is a prospective study, single center based, case-control study. Authors contain polysomnographic data of OSA children and evaluated microbiome in 67 children, which is relatively large number. The results are interesting and valuable. However, specific study design need to be compensated.

REPLY. Thank you for helping us improve our work. Your comments are genuinely appreciated. We’ve revised the manuscript accordingly, and hopefully it will meet your expectations.

 -Introduction:

In line 52, 67, authors described that ‘adenotonsillar’ hypertrophy and ‘adenotonsillar’ microbiome are important in OSA. However, from line 70, authors only comment on ‘tonsillar’ microbiome. Why did authors only consider tonsillar microbiome in OSA patients

REPLY. Thank you for pointing this out. Kim and Min found that the microbiome was not significantly different between the adenoids and tonsils at the phylum level, and the alpha and beta indices were likewise not significantly different between these two regions (doi: 10.21053/ceo.2020.01634). The tonsil size is one of the most important predictors for apnea-hypopnea index (AHI) in preschoolers and school-age children (doi:10.3390/ijerph17134663). Therefore, the influence of the tonsil microbiome may be significant on OSA in young children. Therefore, we only considered tonsillar microbiome in the present study. We have revised the Introduction with adding this information, thanks!

Modified text, Page 1, Lines 68-71

‘… and heat shock protein 27 [23].

Two main methods are used to detect bacterial communities on tonsils …’

-->

‘… and heat shock protein 27 [23].

Tonsil size is one of the most important predictors for apnea-hypopnea index (AHI) in preschoolers and school-age children [24]. Therefore, the influence of the tonsil micro-biome may be significant on OSA in young children. Two main methods are used to detect bacterial communities on tonsils …’

 -Materials and Methods:

How was ‘chronic tonsillitis’ diagnosed?

REPLY.

Thank you for the question. Chronic tonsillitis was identified with symptoms of tonsillitis for a period longer than three months (doi: 10.1002/14651858.CD001802.pub3). We have added this information in the Materials and Methods in the revised manuscript.

Modified text, Page 3, Lines 107-116

‘… were matched by the proportion of chronic tonsillitis. Fig. 1 shows the flow diagram of the study.’

-->

‘… were matched by the proportion of chronic tonsillitis. Both groups were matched by the proportion of chronic tonsillitis. Chronic tonsillitis was defined as symptoms of tonsillitis that persisted for a period longer than three months [31]. … Fig. 1 shows the flow diagram of the study.’

 Why did authors include underweight children together with overweight children into a same group? The pathophysiology of overweight and underweight in OSA may be too different. They should be considered in a different group.

REPLY.

Thank you very much for this in-depth question. In our original consideration, we hypothesized that the tonsil microbiome might be associated with the weight status and anthropometrics of pediatric OSA patients. We recruited consecutive pediatric patients with OSA to participate in this study, used a convenience sampling, and classified them into “non-healthy-weight group and “healthy-weight” group. There were 30 children with overweight and 3 children with underweight. We fully agreed that the pathophysiology of these two subgroups might be too different. Therefore, we further compared variables of interest and found that the differences in patient characteristics, polysomnography variables, blood pressures, and tonsil microbiome did not reach a statistical significance. Accordingly, we have provided these comparisons in the revised Results and Supplementary Materials.

Modified text, Page 6, Lines 230-233

Add:

-->

‘Furthermore, the differences in participants’ characteristics, PSG variables, and BP between the overweight and underweight subgroups were not statistically significant (Supplementary Table S1). These variables were also comparable across the overweight, underweight, and healthy-weight subgroups (all p > 0.05).’

 Modified text, Page 6, Lines 248-255

‘… However, there were no significant differences in α diversity, β diversity, and relative abundances of the top 10 tonsil families between the two subgroups (Figures 2D–2F; all p > 0.05).’

-->

‘… However, there were no significant differences in α diversity, β diversity, and relative abundances of the top 10 tonsil families between the non-healthy-weight and healthy-weight subgroups (Figures 2D–2F; all p > 0.05). Furthermore, the α diversity, β diversity, and relative abundance of the top 10 tonsil families of the over-weight were comparable with those of the under-weight subgroups (p = 0.064, 0.106, 0.492, respectively). Moreover, the differences in α diversity, β diversity, and relative abundances of the top 10 tonsil families across the overweight, underweight, and healthy-weight subgroups (p = 0.088, 0.119, 0.700, respectively).’

 Modified text, Page 7, Lines 261-267

‘… Figures 3A and 3B demonstrate the similar distributions of the relative abundances of these phyla in the non-healthy-weight and healthy-weight subgroups (p > 0.05).’

-->

‘… Figures 3A and 3B demonstrate the similar distributions of the relative abundances of these phyla in the non-healthy-weight and healthy-weight subgroups (p > 0.05). Furthermore, the relative abundances of these phyla in the over-weight and underweight subgroups were comparable (p = 0.536). Additionally, the differences in the 10 most common phyla across the overweight, underweight, and healthy-weight subgroups were not statistically significant (p = 0.814).’

How about the time schedule for polysomnography and tonsillectomy? (For example, polysomnography was performed at ---- weeks before surgery…)

REPLY.

Thank you for helping us present our protocol with better clarity. The median time interval between polysomnography and adenotonsillectomy was ten weeks (IQR: 6–19 weeks). We have revised the Results with this information added.

Modified text, Page 5, lines 215-216

‘… All participants underwent adenotonsillectomy and were included for primary statistical analysis.’

-->

‘… All participants underwent adenotonsillectomy and were included for primary statistical analysis. The median time interval between PSG and adenotonsillectomy was ten weeks (IQR: 6–19 weeks).’

Information about medication history also should be described in study (including antibiotics).

REPLY. Thank you very much for this important comment. We have revised the Materials and Methods with the information added.

Modified text, Page 3, Lines 108-116

‘… were matched by the proportion of chronic tonsillitis. Fig. 1 shows the flow diagram of the study.’

-->

‘… were matched by the proportion of chronic tonsillitis. … Patients with acute inflammation, such as rhinosinusitis, tonsillitis, gastrointestinal infection, or other conditions that needed antibiotic treatment, did not undergo surgery after the diseases diminished for at least two weeks [15]. … Fig. 1 shows the flow diagram of the study.’

 Not only ‘chronic inflammation’, but ‘acute inflammation’ such as acute rhinosinusitis is a confounding factor in this study, and should be evaluated based on objective evaluation such as PNS X-ray.

REPLY.

Thank you very much for the comment. We fully agree that acute inflammation, such as rhinosinusitis, tonsillitis, gastrointestinal infection, or other conditions that needed antibiotic treatment, is a confounding factor in this study and is also a relative contra-indication for adenotonsillectomy. We have revised the Materials and Methods as mentioned above.

Modified text, Page 3, Lines 108-116

‘… were matched by the proportion of chronic tonsillitis. Fig. 1 shows the flow diagram of the study.’

-->

‘… were matched by the proportion of chronic tonsillitis. … Patients with acute inflammation, such as rhinosinusitis, tonsillitis, gastrointestinal infection, or other conditions that needed antibiotic treatment, did not undergo surgery after the diseases diminished for at least two weeks [15]. … Fig. 1 shows the flow diagram of the study.’

Subjective discomfort could be also an important contributing factor. Did authors collect any data about subjective discomfort score from patients or parents?

REPLY. Thank you very much for this cogent comment. We have evaluated OSA symptoms by using the Chinese version of the OSA-18 questionnaire to evaluate their quality of life (doi:10.1111/pcn.12331). We have revised the Materials and Methods and Results with the information added.

Modified text, Page 3, Lines 108-116

‘… were matched by the proportion of chronic tonsillitis. Fig. 1 shows the flow diagram of the study.’

-->

‘… were matched by the proportion of chronic tonsillitis. … Subjective OSA symptoms (evaluated using the Chinese version of the OSA-18 questionnaire [32, 33]), … Fig. 1 shows the flow diagram of the study.’

Modified text, Page 6, Table 1

Add:

OSA-18 score

80 (69‒92)

81 (70‒91)

0.928

Modified text, Page 5, Lines 222-225

‘… Furthermore, there were no statistically significant differences in the proportion of male sex, AHI, RDI, N1 stage, N2 stage, N3 stage, rapid eye movement (REM) stage, and DBP percentile. …’

-->

‘… Furthermore, there were no statistically significant differences in the proportions of male sex … , OSA-18 score, AHI, RDI, N1 stage, N2 stage, N3 stage, rapid eye movement (REM) stage, and DBP percentile. …’

 Regarding tissue sampling, ‘tonsils with crytps’ means superficial? or deep tissue biopsy? Microbiome between superficial area and deep tissue could be different, and tissue sampling is an important issue in microbiome study. And it would be better to add information about the size of samples.

REPLY. Thank you very much for these highly professional comments. Because bacterial colonies were mainly observed in the tonsil crypts rather than in the tonsil follicles (doi: 10.1016/j.ijporl.2018.07.041), we used the superficial tonsils with crypts for molecular examinations. Genomic DNA was immediately extracted from the superficially biopsied specimens (3 mm × 3 mm × 3 mm). We have revised the introduction with the information added.

Modified text, Page 4, lines 147-155

‘Tonsils with crypts were excised using sterile scissors during adenotonsillectomy. The specimens were rinsed with normal saline to remove superficial debris several times after harvesting. Genomic DNA was immediately extracted from the specimens using an EasyPrep Genomic DNA Extraction Kit …’

-->

‘The current study chose tonsil as the primary site for investigation since previous research suggested that adenoidal microbiome was compatible with the tonsillar microbiome at phylum level [23]. Also, since bacterial colonies were mainly observed in the tonsil crypts rather than in the tonsil follicles [19], the superficial tonsils with crypts were used for molecular examinations.

Tonsils with crypts were excised using sterile scissors during adenotonsillectomy. The specimens were rinsed with normal saline to remove superficial debris several times after harvesting. Genomic DNA was immediately extracted from the superficially biopsied specimens (3 mm × 3 mm × 3 mm) using an EasyPrep Genomic DNA Extraction Kit …’

 -Results

In table 1, information about tonsil size should be added. Also the difference of tonsil size between chronic tonsillitis and without chronic tonsillitis also should be evaluated. Size of tonsil and adenoid is an important contributing factor in hypoxia event of children.

REPLY.

Thank you very much for these valuable comments. We fully agreed that the sizes of tonsil and adenoid are important contributing factors in the hypoxia events of children. We have added the information regarding the tonsil size and adenoid-nasopharyngeal ratio (ANR) in the revised Materials and Methods and Results.

We found that the differences in the tonsils size and ANR between the non-healthy-weight and healthy-weight groups were not statistically significant. Furthermore, the difference in the tonsil size between children with chronic tonsillitis (3 [IQR, 3–3]) and children without chronic tonsillitis (3 [IQR, 3–4]) did not reach a statistical significance (p = 0.13). We have revised the Results with the information added.

Modified text, Page 3, lines 108-116

‘… Both groups were matched by the proportion of chronic tonsillitis. Fig. 1 shows the flow diagram of the study.’

-->

‘… Both groups were matched by the proportion of chronic tonsillitis. … Subjective OSA symptoms (evaluated using the Chinese version of the OSA-18 questionnaire [32, 33]), tonsil size (rated using the Brodsky grading scale [34]), the adenoidal-nasopharyngeal ratio (ANR) (measured using lateral radiography of the nasopharynx [35]), and allergic rhinitis were recorded. Fig. 1 shows the flow diagram of the study.’

 Modified text, Page 5-6, lines 222-229

‘… Furthermore, there were no statistically significant difference in the proportions of male sex, AHI, RDI, N1 stage, N2 … than the healthy-weight group.’

-->

‘… Furthermore, there were no statistically significant differences in the proportions of male sex and allergic rhinitis, tonsil size, ANR, OSA-18 score, AHI, RDI, N1 stage, N2 … than the healthy-weight group. Difference in the tonsil size between children with chronic tonsillitis (3 [IQR, 3–3]) and children without chronic tonsillitis (3 [IQR, 3–4]) did not reach a statistical significance (p = 0.13).’

 Modified text, Page 6, Table 1

Add:

Tonsil size

3 (3‒4)

2 (3‒4)

0.461

ANR

0.73 (0.62‒0.83)

0.81 (0.72‒0.87)

0.053

 Authors grouped over-weight and under-weight patients into a ‘non-healthy weight subgroup’. How about the results when only over-weight or underweight patients were evaluated?

REPLY.

Thank you for this valuable question. We have compared variables of interest between the over-weight subgroup (n = 30) and the under-weight subgroup (n = 3) or across the three weight subgroups in the revised Results. We have found that the differences in patient characteristics, polysomnography variables, blood pressures, and tonsil microbiome did not reach a statistical significance. These comparisons were added in the revised Results and Supplementary Materials.

Modified text, Page 6, Lines 230-233

Add:

-->

‘Furthermore, the differences in participants’ characteristics, PSG variables, and BP between the over-weight and under-weight subgroups were not statistically significant (Supplementary Table S1). Moreover, these variables were comparable across the overweight, underweight, and healthy-weight subgroups (all p > 0.05).’

 Modified text, Page 6, Lines 248-255

‘… However, there were no significant differences in α diversity, β diversity, and relative abundances of the top 10 tonsil families between the two subgroups (Figures 2D–2F; all p > 0.05).’

-->

‘… However, there were no significant differences in α diversity, β diversity, and relative abundances of the top 10 tonsil families between the non-healthy-weight and healthy-weight subgroups (Figures 2D–2F; all p > 0.05). Furthermore, the α diversity, β diversity, and relative abundance of the top 10 tonsil families of the overweight were comparable with those of the underweight subgroups (p = 0.064, 0.106, 0.492, respectively). Moreover, the differences in α diversity, β diversity, and relative abundances of the top 10 tonsil families across the overweight, underweight, and healthy-weight subgroups (p = 0.088, 0.119, 0.700, respectively).’

 Modified text, Page 7, Lines 261-267

‘… Figures 3A and 3B demonstrate the similar distributions of the relative abundances of these phyla in the non-healthy-weight and healthy-weight subgroups (p > 0.05).’

-->

‘… Figures 3A and 3B demonstrate the similar distributions of the relative abundances of these phyla in the non-healthy-weight and healthy-weight subgroups (p > 0.05). Furthermore, the relative abundances of these phyla in the overweight and underweight subgroups were comparable (p = 0.536). Additionally, the differences in the 10 most common phyla across the overweight, underweight, and healthy-weight subgroups were not statistically significant (p = 0.814).’

 Modified text, Page 12, Lines 440-442

Add:

-->

Supplementary Materials: The following are available online at www.mdpi.com/xxx/s1, Table S1: Patient characteristics, polysomnography variables, and blood pressures of the overweight and underweight subgroups.’

 Supplementary Materials: Add Table S1.

Table S1. Patient characteristics, polysomnography variables, and blood pressures of the over-weight and under-weight subgroups.

Variables

Over-weight subgroup

Under-weight subgroup

p-Value1

n = 30

n = 3

Patient characteristics

Age (years)

7.0 (6.0‒10.0)

8.0 (5.0‒)

> 0.999

Male sex, n (%)

25 (83%)

3 (100%)

> 0.999

Chronic tonsillitis, n (%)

6 (20%)

0 (0%)

> 0.999

Allergic rhinitis, n (%)

19 (63%)

3 (100%)

0.534

BMI (kg/m2) z-score

2.10 (1.40‒2.44)

-7.60 (-7.45‒)

< 0.001 *

Tonsil size

3 (3‒4)

4 (3‒)

0.260

ANR

0.73 (0.63‒0.83)

0.74 (0.67‒)

0.571

OSA-18 score

80 (69‒94)

77 (73‒)

0.614

Polysomnography variables

AHI (events/h)

9.6 (5.1‒26.3)

6.1 (6.0‒)

0.571

RDI (events/h)

12.1 (7.3‒29.7

13.1 (6.0‒)

0.791

ODI (events/h)

8.5 (3.4‒23.2)

3.2 (1.8‒)

0.100

Mean SpO2 (%)

97 (96‒98)

92 (91‒)

0.082

Minimal SpO2 (%)

89 (83‒91)

91 (88‒93)

0.070

N1 stage

14 (6‒21)

10 (9‒)

0.837

N2 stage

37 (31‒45)

41 (37‒)

0.491

N3 stage

28 (23‒31)

19 (17‒)

0.260

REM stage

18 (12‒22)

21 (21‒)

0.149

Blood pressure variables

Systolic BP, mmHg

111 (100‒121)

85 (61‒)

0.188

Diastolic BP, mmHg

66 (61‒74)

64 (46‒)

0.571

Systolic BP percentile (%)

85 (66‒91)

17 (8‒)

0.260

Diastolic BP percentile (%)

75 (55‒87)

71 (27‒)

0.701

Note: Data are summarized as median (interquartile range) or n (%) as appropriate. Abbreviations: AHI, apnea–hypopnea index; ANR, adenoidal-nasopharyngeal ratio; BMI, body mass index; BP, blood pressure; ODI, oxygen desaturation index; OSA, obstructive sleep apnea; RDI, respiratory disturbance index; REM, rapid eye movement; SpO2, pulse oxygen saturation. 1 Data were compared using the Mann-Whitney U test for continuous variables, and the chi-square test for categorical variables. * Significant differences p < 0.05.

 -Discussion

Again, the results not between’ non-healthy’ and ‘healthy’ but between ‘overweight’ and ‘normal-weight’ or ‘underweight’ and ‘overweight’ might be more convincing, and need to be mentioned here.

REPLY.

We fully agree this valuable comment. We’ve added comparisons on variables of interest between the over-weight subgroup (n = 30) and the under-weight subgroup (n = 3) or across the three weight subgroups in the revised Results. The patient number of the under-weight subgroup was too small to make a conclusion though. We’ve addressed this issue in the study limitation as well.

Modified text, Page 12, lines 416-419

‘Several limitations should be addressed in this study. First, the study cohort was predominantly’

-->

‘Several limitations should be addressed in this study. First, the patient number of the under-weight subgroup was too small and insufficient to make a conclusion. Future studies with a larger sample size of each weigh status subgroup are warranted to further understand how obesity or underweight may impact on the tonsil microbiome and its interaction with OSA. Second, the study cohort was predominantly …’

 To further support authors suggestion (for examples, significant association between intermittent hypoxia and Bacteroidetes), chronic intermittent hypoxia mouse model would be helpful in further study.

REPLY.

Thank you very much for this in-depth comment. We have revised the Discussion with the information added.

Modified text, Page 12, lines 413-415

‘… Intermittent hypoxemia (in terms of mean SpO2 ≤ 97%) further enhances the survival advantage of Bacteroidetes relative to other aerobes of the tonsils.’

-->

‘… Intermittent hypoxemia (in terms of mean SpO2 ≤ 97%) further enhances the survival advantage of Bacteroidetes relative to other aerobes of the tonsils. Mouse models of chronic intermittent hypoxia would be helpful to further validate these inferences [62].’

 Allergic condition is an important factor in regional microbiome. If authors have data about allergic diseases such as allergic rhinitis, allergic asthma, it should be added in table 1 and it should be considered in data analysis.

REPLY.

Thank you very much for these valuable comments. We have found the difference in the proportion of allergic rhinitis between the two subgroups was not statistically significant. However, we have excluded patients with chronic inflammatory disorders (including atopic dermatitis, allergic asthma, and autoimmune disorders) from this study. We have revised Materials and Methods and Results to present our protocol with more clarity.

Modified text, Page 3, lines 103-105

‘… Patients with craniofacial, neuromuscular, or chronic inflammatory disorders were excluded [14, 27]. …’

-->

‘… Patients with craniofacial, neuromuscular, or chronic inflammatory disorders (such as atopic dermatitis, asthma, or autoimmune disease) were excluded [14, 27]. …’

Modified text, Page 3, lines 108-116

‘… Both groups were matched by the proportion of chronic tonsillitis. Fig. 1 shows the flow diagram of the study.’

-->

‘… Both groups were matched by the proportion of chronic tonsillitis. … Subjective OSA symptoms (evaluated using the Chinese version of the OSA-18 questionnaire [32, 33]), …, and allergic rhinitis were recorded. Fig. 1 shows the flow diagram of the study.’

 Modified text, Pages 5, lines 223-225

‘… Furthermore, there were no statistically significant difference in the proportions of male sex, AHI, RDI, N1 stage, N2 … and DBP percentile. …’

-->

‘… Furthermore, there were no statistically significant differences in the proportions of male sex and allergic rhinitis, … and DBP percentile. …’

 Modified text, Page 6, Table 1

Add:

Allergic rhinitis, n (%)

22 (67%)

25 (76%)

0.587

Reviewer 2 Report

The article contains original data, the relationship between the microbiota and OSA needs to be further explored.

Author Response

Reviewer 2’s comments:

The article contains original data, the relationship between the microbiota and OSA needs to be further explored.

REPLY.

REPLY. We appreciate these encouraging comments. We have amended the modified text according your suggestions. Hopefully the quality of this revised manuscript will meet your expectations. Thank you for helping us improve our work!

Modified text, Page 12, lines 425-428

‘… Third, the study was cross-sectional and thus unable to conclude the direction of associations or causal effects. Future prospective investigations on the effects of OSA treatment on the tonsil microbiome with a larger sample size will be of interest.’

-->

‘… Forth, the study was cross-sectional and thus unable to conclude the direction of associations or causal effects. The relationships between the microbiota and OSA needs to be further explored. Also, future prospective investigations on the effects of OSA treatment on the tonsil microbiome with a larger sample size will be of interest.’

Round 2

Reviewer 1 Report

Authors tried to response to reviewer's comments specifically.

The manuscript is quite improved.